# Fine Mapping and Identification of a Candidate Gene for the Glossy Green Trait in Cabbage (*Brassica oleracea* var. *capitata*)

**DOI:** 10.3390/plants12183340

**Published:** 2023-09-21

**Authors:** Peiwen Wang, Ziheng Li, Lin Zhu, Mozhen Cheng, Xiuling Chen, Aoxue Wang, Chao Wang, Xiaoxuan Zhang

**Affiliations:** 1Key Laboratory of Biology and Genetic Improvement of Horticultural Crops (Northeast Region), Ministry of Agriculture and Rural Affairs, Northeast Agricultural University, Harbin 150030, China; wangpeiwen@neau.edu.cn (P.W.); lzh18236452381@163.com (Z.L.); 18645096256@163.com (L.Z.); mzcheng@neau.edu.cn (M.C.); chenx@neau.edu.cn (X.C.); axwang@neau.edu.cn (A.W.); wangchao504@126.com (C.W.); 2College of Horticulture and Landscape Architecture, Northeast Agricultural University, Harbin 150030, China

**Keywords:** cuticular wax, cabbage, genetic mapping, BSA-seq, Agenet/Tudor domain

## Abstract

In higher plants, cuticular wax deposited on the surface of epidermal cells plays an important role in protecting the plant from biotic and abiotic stresses; however, the molecular mechanism of cuticular wax production is not completely understood. In this study, we identified a glossy green mutant (98-1030gl) from the glaucous cabbage inbred line 98-1030. Scanning electron microscopy indicated that the amount of leaf cuticular wax significantly decreased in 98-1030gl. Genetic analysis showed that the glossy green trait was controlled by a single recessive gene. Bulked segregant analysis coupled with whole genome sequencing revealed that the candidate gene for the glossy green trait was located at 13,860,000–25,070,000 bp (11.21 Mb) on Chromosome 5. Based on the resequencing data of two parents and the F2 population, insertion-deletion markers were developed and used to reduce the candidate mapping region. The candidate gene (*Bol026949*) was then mapped in a 50.97 kb interval. Bol026949 belongs to the Agenet/Tudor domain protein family, whose members are predicted to be involved in chromatin remodeling and RNA transcription. Sequence analysis showed that a single nucleotide polymorphism mutation (C → G) in the second exon of *Bol026949* could result in the premature termination of its protein translation in 98-1030gl. Phylogenetic analysis showed that *Bol026949* is relatively conserved in cruciferous plants. Transcriptome profiling indicated that *Bol026949* might participate in cuticular wax production by regulating the transcript levels of genes involved in the post-translational cellular process and phytohormone signaling. Our findings provide an important clue for dissecting the regulatory mechanisms of cuticular wax production in cruciferous crops.

## 1. Introduction

Cuticular wax is a diverse class of C_20_–C_34_ very-long-chain fatty acids (VLCFAs) and their derivatives, including aldehydes, alcohols, alkanes, ketones, and wax esters. Cuticular wax overlays and blends in cutin consisting of fatty acid monomers, including C_16_ and C_18_ hydroxy, epoxy, and dicarboxylic fatty acids. Cuticular wax and cutin together form the cuticle in the outermost layer of land plants. Based on the location of the cuticular wax, it is divided into two types: epicuticular wax covering cutin polymers and intra-cuticular wax embedding cutin polymers [1]. The biosynthesis of cuticular wax originates from a de novo synthesis of C_16_ or C_18_ fatty acids, which are then catalyzed to form C_16_ or C_18_ acyl-CoAs by a long-chain acyl-CoA synthase in plastids [1,2]. The resulting C_16_ or C_18_ acyl-CoAs are subsequently exported into the endoplasmic reticulum (ER) and undergo a series of enzyme catalysis reactions, in which a C_2_ unit is successively added into C_16_ or C_18_ acyl-CoAs to generate VLCFAs-CoAs by the multienzyme fatty acid elongase [1,2]. VLCFAs-CoA derivatives, including secondary alcohols, aldehydes, alkanes, and ketones, are produced in the alkane-forming pathway, and other VLCFAs-CoA derivatives, including primary alcohols and wax esters, are yielded in the alcohol-forming pathway [1,2]. After wax components are synthesized in the ER, they are then delivered to the cuticle through the plasma membrane and cell wall, finally depositing on the plant surface [1,2].

A large number of reports have demonstrated that cuticular wax plays an important role in resistance to biotic and abiotic stresses, such as pathogen infection [3], insect attack [4], drought [5], low temperature [6], and UV irradiance [7]. Due to the importance of cuticular wax in plants, numerous critical genes involved in the biosynthesis, translocation, and regulation of cuticular wax have been extensively characterized in many species, including *Arabidopsis* [8,9,10], *Oryza sativa* [11], *Zea mays* [12], and *Solanum lycopersicum* [13]. Moreover, some cuticular wax biosynthesis-related genes are also identified in many cruciferous crops, including *Brassica napus* [14], *Brassica rapa* [15], and *Brassica oleracea* [16]. However, the regulatory mechanism of cuticular wax production remains poorly understood in cruciferous crops.

Cabbage (*Brassica oleracea* var. *capitata*) is one of the most popular vegetable crops throughout the world and becomes a model for the study of functional genes in cruciferous crops because of the relatively simple structure of its diploid genome, which is approximately 630 Mb in size, with nine pairs of chromosomes (2n = 2x = 18). In recent decades, with the rapid development of DNA sequencing technology, a high-quality reference genome sequence of cabbage has been provided [17], which remarkably promotes the mapping and identification of candidate genes controlling many important biological processes in cabbage based on insertion-deletion (InDel) variation and single nucleotide polymorphism (SNP), such as hybrid lethality [18], flowering [19], and black rot resistance [20]. Furthermore, due to the lack of wax crystals covering the plant surface, numerous wax-deficient cabbage mutants caused by spontaneous gene mutations generally exhibit the glossy green trait compared to glaucous wild-type plants [16,21,22], which not only makes it possible to easily observe phenotypes in genetic analysis but also provides many valuable genetic recourses for the study on cuticular wax production. These two advantages greatly facilitate the unraveling of the regulatory mechanism of cuticular wax production in cabbage using wax-deficient mutants and molecular markers.

In the present study, a glossy green mutant (98-1030gl) in the background of the normal cabbage inbred line 98-1030 was characterized and analyzed. Inheritance of the glossy green trait in an F_2_ population was investigated by crossing 98-1030gl and 98-1030. The candidate gene controlling the glossy green trait was located in an 11.21 Mb region on Chromosome 5 via bulked segregant analysis coupled with whole genome sequencing (BSA-seq). The linkage analysis and fine mapping of the candidate gene for the glossy green trait were performed using InDel markers developed from InDel variations. The candidate gene (*Bol026949*) encoding an Agenet/Tudor domain protein was found in a 50.97 kb region. Sequence analysis showed that C at the 711th position was replaced by G in the coding sequence of *Bol026949* in 98-1030gl, thereby leading to the premature formation of a stop codon. RNA-seq analysis indicated that *Bol026949* might affect the transcript levels of genes related to the post-translational cellular process and phytohormone signaling. Our achievements provide a deeper insight into the regulatory mechanisms of cuticular wax production in cruciferous crops.

## 2. Results

### 2.1. Obtaining the Homozygous Glossy Green Mutant

The glossy green mutant was initially discovered in the inbred line 98-1030 with a glaucous appearance. Plants with a glossy green appearance were selected and self-pollinated for 10 generations to obtain the homozygous glossy green mutant. Finally, a glossy green population (98-1030gl), whose progeny individuals all exhibited a glossy green appearance, was collected, indicating that the homozygous glossy green mutant was obtained.

### 2.2. Phenotypic Characterization of the Glossy Green Mutant and Its Wild Type

Cotyledons, true leaves at the five-leaf stage, and leaves at the heading stage were harvested and observed to analyze the phenotypic differences between the glossy green mutant (98-1030gl) and its wild type (WT). As shown in Figure 1A, there were no obvious differences between 98-1030gl and WT cotyledons. True leaves from 98-1030gl at the five-leaf stage displayed a glossy green appearance compared to their wild type (Figure 1B). At the heading stage, cabbage leaves from 98-1030gl exhibited a significant glossy green appearance, whereas WT leaves remained glaucous (Figure 1C,D).

### 2.3. Observation of Wax Crystal on the Surface of 98-1030gl and WT Leaves

Given the importance of the coverage of cuticular wax in cabbage leaf appearance, we further observed wax crystals on the surface of leaves. The results indicated that wax crystals were not generated in 98-1030gl and WT cotyledons (Figure 2A,B). At the five-leaf stage, scale-like wax crystals accumulated on the surface of WT leaves, but not on those of 98-1030gl (Figure 2C,D). A large number of compact wax crystals emerged in WT leaves at the heading stage, whereas a small number of granular wax crystals dispersed on the surface of 98-1030gl leaves (Figure 2E,F). These results are exactly consistent with the observation of cabbage leaf appearance, suggesting that the failure of cuticular wax deposition is responsible for the glossy green trait of 98-1030gl leaves.

### 2.4. Genetic Analysis of the Glossy Green Trait

A six-generation genetic population was constructed by crossing 98-1030gl with 98-1030 and backcrossing F_1_ plants with the parents to understand the inheritance of the glossy green trait. All 132 individuals in the F_1_ population showed a glaucous appearance (Table 1). In the F_2_ population, 650 individuals had glaucous leaves, and 225 individuals exhibited a glossy green appearance (Table 1). The segregation ratio was approximately 3:1 by the Chi-square test (χ^2^ = 0.22 < χ^2^_0.05_ = 3.84). Furthermore, all 204 BC_1_ individuals possessed glaucous leaves, and individuals in the BC_2_ population included 92 plants with glaucous leaves and 76 plants with glossy leaves, respectively (Table 1), corresponding to a ratio of 1:1 by the Chi-square test (χ^2^ = 1.52 < χ^2^_0.05_ = 3.84). These results collectively suggest that the glossy green trait is controlled by a single recessive gene.

### 2.5. Fine Mapping of the Candidate Gene

BSA-seq was performed in the parents and F_2_ population to locate the candidate gene for the glossy green trait. After resequencing, 75.21 Mb and 70.94 Mb of clean reads were obtained from 98-1030 and 98-1030gl, respectively, while an average of 101.35 Mb of clean reads was obtained from the F_2_ plants (Appendix A). The mean values of Q30 and GC were 91.92% and 37.90%, respectively (Appendix A). All sequencing data of the parents and F_2_ population were mapped in the reference genome. The mapping rate of the parents and F_2_ population ranged from 96.98% to 97.20%. The average sequencing depth of the parents was >16×, and the mapping rate was >85% (at least 1×). The average sequencing depth of the F_2_ population was >22×, and the mapping rate was >86% (at least 1×) (Appendix A). Based on the resequencing results above, SNPs were detected and filtered using the genome analysis toolkit (GATK) software v4.1.4.0. Finally, a total of 3,365,031 high-quality confident SNPs and 808,175 high-quality confident InDels were identified in the parents and F_2_ population (Appendix A). The ΔSNP index (the confidence value was 99%) algorithm was used to analyze the potential position of the candidate gene. The results showed that the candidate gene for the glossy green trait was mapped to an 11.21 Mb region from 13,860,000 to 25,070,000 bp on Chromosome 5 (Figure 3A).

To fine-map the chromosomal region of the candidate gene, 4149 InDel primers were developed based on the preliminary mapping region and resequencing results, and 384 InDel primers were randomly selected to screen polymorphic primers. A total of 84 primers exhibited polymorphism between the parents, and then 14 of them were selected and used to screen the F_2_ population (875 individuals) with 16 recombinant individuals. The results showed that the candidate gene was mapped in a 539.43 kb interval between ND78 and ND111 (15,649,101–16,188,528 bp) (Figure 3B). To further narrow down the potential region of the candidate gene, 32 InDel markers were designed in the interval of ND78 and ND111 based on the resequencing data, and 8 of them showed polymorphism between the parents. These 8 co-dominant polymorphic markers were chosen and used to screen a bigger F_2_ population (2252 individuals) with 11 recombinant individuals. The results showed that the candidate gene was located in the region between ND78-3 and ND78-4, with a physical distance of 50.97 kb (15,766,620–15,817,588 bp) (Figure 3B).

### 2.6. Identification of the Candidate Gene

Based on the cabbage genome database, there were five genes in the mapped region (Table 2). Gene function annotation and sequence analysis showed that three of the five genes had definite molecular functions, including Agenet domain-containing protein (*Bol026949*), plastocyanin (*Bol026950*), and tRNA pseudouridylate synthase (*Bol026952*). Among them, only *Bol026949* had a nonsynonymous mutation in its coding region, indicating that *Bol026949* could be the candidate gene for the glossy green trait. Bol026949 belongs to the Agenet/Tudor domain protein family (Table 2), whose members are predicted to be involved in chromatin remodeling and RNA transcription [23]. Sequence alignment showed that C was replaced by G at the 15,777,188 bp in the second exon region of *Bol026949* in 98-1030gl, thereby leading to the formation of a stop codon (TAG) (Figure 4A). This mutation was predicted to cause the premature termination of Bol026949 protein translation, thus resulting in the loss of domains at its C-terminus (Figure 4B). The genomic sequences of *Bol026949* in 98-1030gl and 98-1030 were analyzed by PCR amplification and sequencing to confirm the SNP mutation in *Bol026949*. The results showed that the SNP variation (C → G) indeed existed in the coding sequence of *Bol026949* in 98-1030gl (Appendix A).

### 2.7. Phylogenic Analysis

The Bol026949 protein sequence was used as a query to retrieve against the protein database in many different species in Phytozome V13.0, using the online BLAST tool to understand the evolution of *Bol026949* in higher plants. Bol026949 and its orthologs were aligned and constructed a neighbor-joining phylogenetic tree with 1000 bootstraps using the MEGA 11.0 software. The results showed that Bol026949 had a close phylogenetic relationship with homologs in cruciferous plants, including *Arabidopsis thaliana*, *Brassica rapa*, and *Crambe hispanica*, indicating that *Bol026949* is relatively conserved in the *Cruciferae* family (Figure 5).

### 2.8. Transcriptome Profiling

Transcriptome profiling of 98-1030gl and WT leaves was performed using RNA-seq to assess the influence of the *Bol026949* mutation on gene expression. As shown in Figure 6A, 3705 differentially expressed genes were confidently identified in 98-1030gl leaves relative to their wild type, including 1793 up-regulated genes and 1912 down-regulated genes (Appendix A). Kyoto Encyclopedia of Genes and Genomes pathway enrichment analysis indicated that these differentially expressed genes were significantly enriched in categories related to metabolic pathways, plant hormone signal transduction, and secondary metabolites biosynthesis (Figure 6B and Appendix A). Unfortunately, these differentially expressed genes were not significantly grouped in categories related to cuticular wax production. To further understand the molecular function of *Bol026949*, clusters of orthologous groups function enrichment analysis of differentially expressed genes was conducted. The results showed that differentially expressed genes were clustered into categories related to post-translational modification, protein turnover, chaperone, and signal transduction mechanism (Figure 6C and Appendix A). These results collectively imply that *Bol026949* may modulate the transcript levels of genes involved in the post-translational cellular process and phytohormone signaling.

## 3. Discussion

Cuticular wax is an essential structure on the surface of higher plants and usually functions as a protective film against external environmental stimuli. Cuticular wax comprises external cuticular wax covering cutin and internal cuticular wax embedded in cutin. In general, external cuticular wax spontaneously forms three-dimensional crystals, which are easily observed by scanning electron microscopy (SEM) [24]. Wax crystals usually have diverse microcharacter in structure and range from 0.5 to 10 μm in height [25]. Plants with wax crystals generally exhibit glaucous phenotype. The amount and composition of wax crystals vary among different species or organisms in the same species. A twofold range in cuticular wax loads has been detected in stems from 40 different *Arabidopsis* ecotypes [26]. Similar results have been observed in 30 different rice cultivars [27]. In *Arabidopsis*, secondary alcohols and ketones predominantly exist in stems, silique walls, flowers, and seed coats, but not on the surface of leaves [2,28,29]. Furthermore, the microscopic conformation, composition, and number of wax crystals also depend on growth, development, and environmental conditions. Zhang et al. have found that the crystal structure of cuticular wax correspondingly changes during grape fruit ripening [30]. Relative air humidity structurally affects wax crystals on the surface of Korla pear fruits [31]. The proportion of aldehyde, primary alcohol, and ester significantly increases in leaf cuticular wax in *Brassica oleracea* var. *gemmifera* under lower light and higher temperature conditions, whereas the biosynthesis of alkanes, secondary alcohol, and ketones is enhanced at higher light and lower temperature levels [32]. The amount of leaf cuticular wax increases approximately twofold in *Arabidopsis* under drought conditions [33]. Barthlott et al. have systematically classified wax crystals according to the morphological structure of cuticular wax in more than 13,000 species and found that wax crystals are mainly platy and tubular and contain 23 different types, including column, bar, flake, line, and chimney [25]. In the present study, cuticular wax in normal cabbage began to form at the stage of five-leaf and showed a sparse scale-like appearance. Wax crystals on the surface of cabbage leaves changed from sparse flakes to dense platelets at the heading stage. This evidence suggests that the biosynthesis of leaf cuticular wax in cabbage initiates at the stage of five-leaf, and its number significantly increases during growth and development. However, little is known about the occurrence mechanism of this phenomenon. Therefore, the molecular mechanisms underlying cuticular wax accumulation on the surface of cabbage leaves during growth and development still deserve to be deeply investigated in the future.

Wax-deficient *Arabidopsis* mutants were reported in 1979 [34], and similar reports were also described for many other species, including rice [35], maize [36], barley [37], and cabbage [16]. In contrast to wild-type plants, the accumulation of glaucous powdery wax on the surface of leaves and stems is completely or partially dampened in wax-deficient mutants, so that wax-deficient leaves and stems generally exhibit a glossy green appearance. In cabbage, numerous wax biosynthesis-related genes have been characterized in glossy green mutants. A large number of genetic analyses have shown that most wax-deficient mutations in cabbage are controlled by a single recessive gene [16,22,38]. Similar to most previous studies, we found that the glossy green trait in the cabbage inbred line 98-1030gl followed a recessive pattern by constructing a six-generation genetic population. In addition to wax-deficient mutations caused by a single recessive gene, a few reports have indicated that the glossy green trait in cabbage adheres to a dominant inheritance. Liu et al. have reported that *BoGL1* is a single dominant gene controlling the glossy green trait in a wax-deficient cabbage mutant [39]. A dominant glossy mutation results in the deficiency of cuticular wax in the cabbage mutant CGL-3 [21]. So far, the identification of the candidate genes for the glossy green trait in model plants has shown that the candidate genes responsible for wax deficiency are predominantly some critical genes involved in wax biosynthesis, transport, and regulation [2]. In this study, we utilized BSA-seq combined with fine mapping to target a candidate gene (*Bol026949*) encoding an Agenet/Tudor domain protein, which was likely to be involved in the accumulation of leaf cuticular wax in cabbage. Sequence analysis revealed that a nucleotide substitution (C → G) in the coding region of *Bol026949* might lead to the premature termination of its protein translation. Based on the functional analysis of homologs in *Arabidopsis*, we speculate that this gene is likely to participate in the regulation of cuticular wax deposition at the transcriptional level.

In higher plants, cuticular wax production is extensively regulated at multiple levels, including transcription, post-transcription, and post-translation [1,2]. Among them, the transcriptional regulation of cuticular wax biosynthesis is the most important mode of regulation. Some critical transcription factors (TFs), including AP2/ERF, MYB, HD-ZIP, and MADS TFs, play a vital role in the transcriptional regulation of cuticular wax biosynthesis [2,40]. An AP2/ERF-type transcription factor WAX INDUCER1/SHINE1 enhances cuticular wax accumulation by up-regulating the expression of wax biosynthesis-related genes [41,42]. Overexpression of MYB96 elevates drought tolerance in *Arabidopsis* plants via the acceleration of cuticular wax biosynthesis [43]. In addition to the above positive regulators, two AP2/ERF-type transcription factors, DEWAX and DEWAX2, have been reported to negatively control cuticular wax biosynthesis by directly targeting wax biosynthesis genes [44,45]. To date, few other types of regulators are reported in the transcriptional regulation of cuticular wax biosynthesis. In this work, we found that *Bol026949* encoding an Agenet/Tudor domain protein seems to be required for cuticular wax production. The Agenet/Tudor protein family is a class of Agenet/Tudor domain-containing proteins across many species, including plants, animals, and humans [23]. In plants, the Agenet/Tudor domain often co-occurs with motifs related to transcriptional, post-translational, and epigenetic regulation, including EMSY-like N-Terminal, Bromo Adjacent Homology, Plant Homeodomain, and Domain of Unknown Function 724 domains [23]. In *Arabidopsis*, the Agenet/Tudor gene family consists of 28 members whose protein products are predicted to be involved in chromatin remodeling and the regulation of gene expression [23]. It is reported that RING-type E3 ubiquitin ligase HUB1 and HUB2 are able to elevate the transcript levels of genes involved in cuticle lipid biosynthesis by monoubiquitinating histone H2B proteins [46], indicating that chromatin remodeling is closely implicated in cuticular wax accumulation [2]. Thus, we analyzed the transcriptome of 98-1030gl and its wild type to further understand how *Bol026949* affects cuticular wax accumulation via transcriptional regulation. Transcriptome profiling showed that differentially expressed genes in 98-1030gl relative to its wild type were significantly enriched in metabolic pathways but rarely participate in wax biosynthesis-related pathways. Furthermore, we also noticed that the expression levels of many genes involved in post-translational regulation and phytohormone signaling exhibited significant alterations. Therefore, we guess that *Bol026949* might be involved in cuticular wax production by modulating the transcript levels of some key post-translational regulators in the phytohormone signaling pathway, rather than by directly affecting the expression of genes involved in cuticular wax biosynthesis. The exact biological function of this gene in cuticular wax production remains to be further investigated by constructing gene knockout and overexpression plants.

## 4. Materials and Methods

### 4.1. Plant Materials and Phenotype Identification

The glossy green mutant 98-1030gl used in this study was initially discovered from the cabbage inbred line 98-1030, which was obtained in our previous study. 98-1030 with a glaucous appearance (P_1_) and 98-1030gl with a glossy green appearance (P_2_) were crossed to generate the F_1_ population. The F_1_ plants were then used to construct the F_2_ population by self-crossing. The F_1_ plants were backcrossed to P_1_ and P_2_ to produce the BC_1_ and BC_2_ populations, respectively. A six-generation genetic population containing the P_1_, P_2_, F_1_, F_2_, BC_1_, and BC_2_ plants was used to analyze the inheritance of the glossy green trait, and the P_1_, P_2_, and F_2_ plants were used for the mapping study of the candidate gene controlling the glossy green trait. All of the plant materials above were derived from the Laboratory of *Brassica oleracea* Genetic and Breeding, College of Horticulture and Landscape, Northeast Agricultural University, and grown in the greenhouse under standard culture conditions (Harbin, China). The phenotypic differences between 98-1030gl and 98-1030 were determined by the color, gloss, and surface texture of cabbage leaves at the five-leaf stage. The segregation ratio of the F_2_ population was validated by the Chi-square test (χ^2^).

### 4.2. SEM Analysis

SEM analysis of wax crystals on the surface of cabbage leaves was performed as previously described by Carvajal et al., with some modifications [47]. In brief, fresh leaves were harvested from cabbage plants at different development stages and then fixed in 2.5% glutaraldehyde overnight. Samples were mounted on a metal stub and coated with gold particles using a sputter-coater. The ultrastructure of wax crystals was observed using a SUPRA-40VP scanning electron microscope (Zeiss, Jena, Germany) equipped with a detector of secondary electron signals at a voltage of 3 kV.

### 4.3. DNA Isolation

The genomic DNAs of two parents (98-1030 and 98-1030gl), F_1_, and F_2_ individuals were extracted from young leaves at the five-leaf stage using the modified cetyltrimethylammonium bromide (CTAB) method [48]. Briefly, 0.2 g of leaves were ground into powder with liquid nitrogen and mixed with the preheated 2% CTAB buffer containing 2% β-mercaptoethanol at 65 °C. The mixture was incubated at 65 °C for 45 min and then added with chloroform/isoamyl alcohol (24:1, *v*/*v*). Subsequently, the mixture was centrifuged at 12,000 rpm for 15 min at 4 °C, and then the supernatant was collected and mixed with chloroform/isoamyl alcohol (24:1, *v*/*v*) again. After centrifugence, the supernatant was mixed with an equal volume of isopropyl alcohol, followed by incubation at −20 °C for 1 h. The precipitated pellets were collected by centrifugation and washed with 75% (*v*/*v*) ethanol twice. The dried pellets were dissolved with deionized H_2_O. The quality and concentration of DNA samples were analyzed by 1.0% agarose gel electrophoresis and a NanoDrop (Thermo Fisher Scientific, Waltham, MA, USA), respectively.

### 4.4. BSA-seq Analysis

For BSA-seq analysis, the genomic DNAs of two parents (98-1030 and 98-1030gl), F_1_, and F_2_ individuals were isolated and adjusted to 100 ng/μL. An equal volume of DNA from 30 individuals with a glaucous appearance and 30 individuals with a glossy green appearance in the F_2_ population was mixed respectively to generate two mixed DNA pools, the glaucous pool (H1-pool) and the glossy pool (H2-pool). Four DNA libraries with 400 bp in insert fragment size, including two parent lines (98-1030 and 98-1030gl), H1-pool, and H2-pool, were constructed for paired-end sequencing analysis on an Illumina HiSeq platform. The resulting raw data were filtered by removing reads with adaptors, reads containing bases with Q ≤ 20, and reads less than 50 bp in size. The filtered reads were aligned to the *Brassica oleracea* var. *capitata* reference genome in Phytozome v13.0 (https://phytozome-next.jgi.doe.gov/ (accessed on 20 May 2020)) using Burrows Wheeler Aligner (BWA, 0.7.12-r1039) with default parameters. All alignment results were converted to SAM/BAM files using the Picard software package v1.107 (http://www.psc.edu/index.php/user-resources/software/picard (accessed on 20 May 2020)). The consistency between all paired-end reads was checked by the Fix Mate Information tool. SNP and InDel variations were analyzed by the GATK program [49]. To improve the accuracy of SNP calling, PCR and optical duplicates were ruled out using the Mark Duplicates tool in the Picard software package v1.107 (http://www.psc.edu/index.php/user-resources/software/picard (accessed on 20 May 2020)), and all reads near InDels were realigned by the Indel Realigner tool in the GATK program [49]. The resulting SNPs and InDels of all four DNA libraries were strictly filtered and then annotated in the reference genome using the ANNOVAR software v2019Oct24 [50]. The SNP index of all SNPs in H1-pool and H2-pool was measured based on the genetic background of the parent line 98-1030. The ΔSNP-index between H2-pool and H1-pool was determined by SNP index differences in H2-pool relative to H1-pool and visualized in the whole genome, which was subsequently used to locate a potential candidate region related to the glossy green trait. The BSA-seq data for the mapping of the candidate gene have been deposited in the Sequence Read Archive (Nos. SRR25906298, SRR25906299, SRR25906300, and SRR25906301).

### 4.5. Marker Development and Fine Mapping

InDel markers were developed and used for fine mapping based on the resequencing data to minimize the region of the candidate gene for the glossy green trait. The primers for InDel markers were designed, and then the polymorphic primers screened based on the parent lines and F_1_ population were used for genotyping F_2_ individuals with the glaucous and glossy appearance. The PCR reaction was performed in a volume of 50 μL mixture (1 × Es Taq Master Mix, 0.4 μM forward or reserve primer, 0.5 μg genomic DNA, and ddH_2_O) with the following programs: initial denaturation at 94 °C for 4 min; amplification by 35 cycles of 94 °C for 30 s, 55 °C for 30 s, and 72 °C for 30 s; and extension at 72 °C for 10 min. The PCR amplicons for InDel markers were separated by 8% polyacrylamide gel electrophoresis at 150 V for 2.5 h and then submitted to silver staining. The polymorphic primers used for fine mapping are listed in Appendix A.

The genotypes of individuals from the F_2_ population were identified using the polymorphic primers for InDel markers described previously to construct the genetic linkage map of the candidate gene locus. Individuals consistent with 98-1030 in genotype were marked as “A”; individuals consistent with 98-1030gl in genotype were marked as “B”; individuals consistent with the F_1_ population in genotype were marked as “H”; and individuals with unknown genotype were marked as “–”. The statistical data associated with InDel markers co-segregated with the candidate gene were analyzed by QTL Icimapping 4.1 [51], and then the linkage analysis was conducted using the Kosambi mapping function in JoinMap 4.0 [52].

### 4.6. Functional Annotation and Sequence Analysis

The functional annotation of genes in the candidate region was performed based on the *Brassica oleracea* var. *capitata* reference genome in Phytozome v13.0 (https://phytozome-next.jgi.doe.gov/ (accessed on 10 March 2023)) and gene ontology enrichment analysis (http://geneontology.org/ (accessed on 10 March 2023)). The specific primers were designed and used to amplify the genomic sequence of the candidate gene from the genomic DNAs of the parent line 98-1030 and 98-1030gl. The resulting PCR products were then sequenced and aligned with the reference genome to validate the SNP mutation identified from BSA-seq analysis. The primers used for the sequence analysis of the candidate gene are listed in Appendix A.

### 4.7. Phylogenetic Analysis

For the phylogenetic analysis of the candidate gene, the sequence alignment of its homologous proteins was performed by Clustal Omega (https://www.ebi.ac.uk/Tools/msa/clustalo/ (accessed on 25 March 2023)) [53]. The alignment results were then submitted to the MEGA 11.0 software to generate a neighbor-joining phylogenetic tree with 1000 bootstrap replicates [54].

### 4.8. RNA-seq

Transcriptome profiling of 98-1030gl and 98-1030 leaves was performed using RNA-seq to analyze differentially expressed genes between 98-1030gl and 98-1030. Total RNA isolation and paired-end sequencing on an Illumina platform were conducted in the Metware Biological Technology Corporation (Wuhan, China). The original data were filtered using fastp, a high-throughput sequence quality control tool [55]. The clean reads were then mapped to the reference genome using HISAT [56]. The FPKM value of each gene was calculated by featureCounts [57]. Differentially expressed genes between 98-1030gl and 98-1030 leaves were selected by DESeq2 [58,59]. The corrected *p* value ≤ 0.05 and |log_2_(fold change)| ≥ 2 were used as thresholds for significant difference expression. Gene function enrichment analysis was conducted based on the hypergeometric test [60,61]. The RNA-seq data for the identification of differentially expressed genes have been deposited in the Sequence Read Archive (Nos. SRR25992310, SRR25992311, SRR25992312, SRR25992313, SRR25992314, and SRR25992315).

## 5. Conclusions

In conclusion, we obtained a wax-deficient mutant from the cabbage inbred line 98-1030. We compared the differences in leaf cuticular wax between them. Inheritance analysis demonstrated that the glossy green trait caused by wax-deficient mutation is controlled by a single recessive gene. Using BSA-seq combined with fine mapping, we located a candidate gene (*Bol026949*) in a 50.97 kb region on Chromosome 5. *Bol026949* encodes an Agenet/Tudor domain protein, which is predicted to participate in chromatin remodeling and gene expression regulation. Sequence analysis revealed that a nucleotide substitution (C → G) in its second exon could lead to the premature termination of protein translation. Phylogenic analysis showed that *Bol026949* is relatively conserved in cruciferous plants. *Bol026949* might be involved in cuticular wax production in cabbage by controlling the transcript levels of genes involved in post-translational regulation and phytohormone signal transduction. Our findings lay a foundation for further unraveling the transcriptional regulatory network of cuticular wax production in cruciferous crops.

## Figures and Tables

**Figure 1 plants-12-03340-f001:**
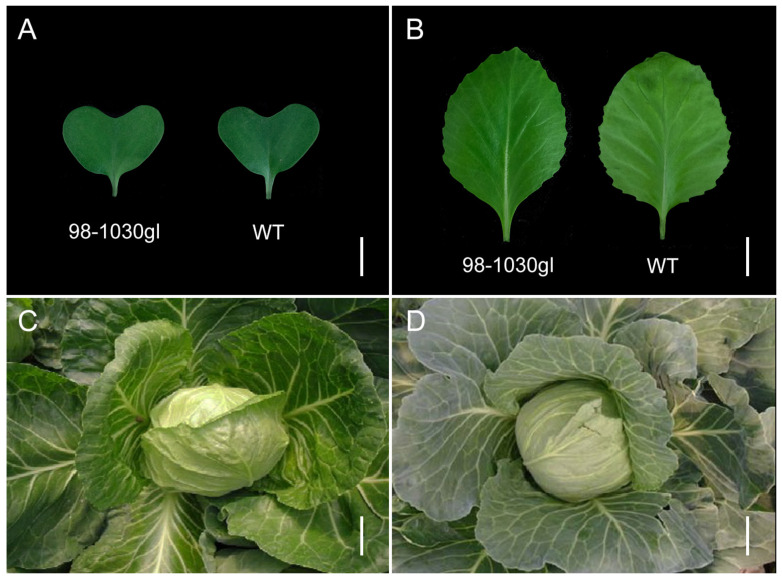
Leaf phenotypes of 98-1030gl and its wild type (WT). (**A**) 98-1030gl and WT cotyledons. Bar, 1 cm. (**B**) 98-1030gl and WT true leaves at the five-leaf stage. Bar, 1 cm. (**C**) 98-1030gl leaves at the heading stage. Bar, 10 cm. (**D**) WT leaves at the heading stage. Bar, 10 cm.

**Figure 2 plants-12-03340-f002:**
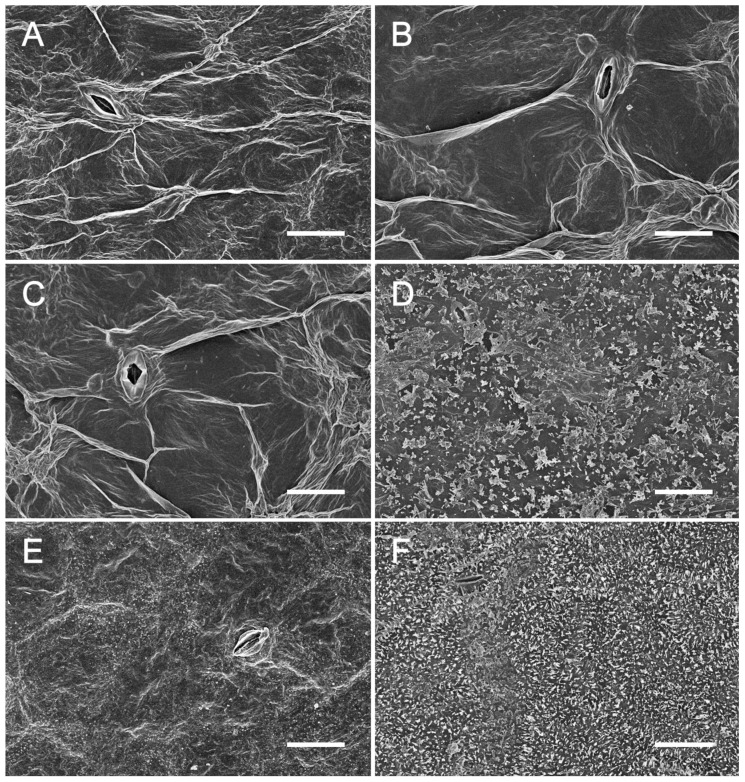
Scanning electron microscopy of cuticular wax on the surface of leaves. (**A**,**B**) Appearance of 98-1030gl (**A**) and WT (**B**) cotyledons. (**C**,**D**) Appearance of 98-1030gl (**C**) and WT (**D**) true leaves at the five-leaf stage. (**E**,**F**) Appearance of 98-1030gl (**E**) and WT (**F**) leaves at the heading stage. Bar, 20 µm.

**Figure 3 plants-12-03340-f003:**
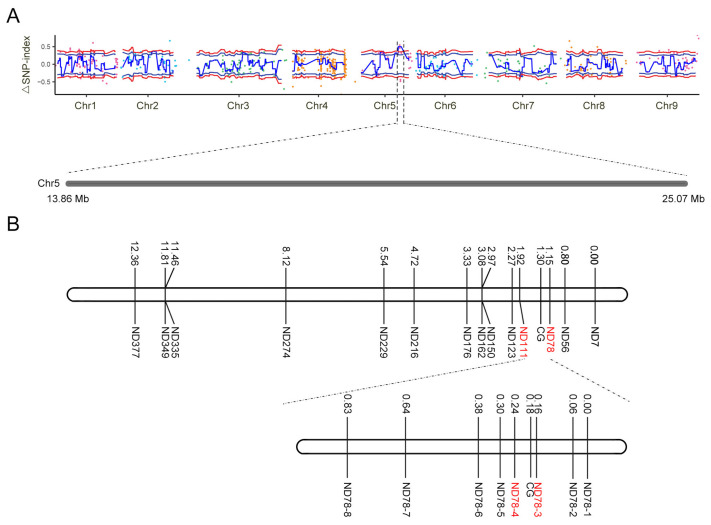
Mapping of the candidate gene controlling the glossy green trait. (**A**) Distribution of ΔSNP-index values on chromosomes (Chr). Chromosome names are shown on the *x*-axis, and colored points represent the ΔSNP-index value of each SNP locus. The blue line indicates the fitted ΔSNP-index value, and the red line indicates the significance association threshold. The candidate gene is mapped in an 11.21 Mb region on Chr5. (**B**) Genetic linkage maps of the candidate gene controlling the glossy green trait. The genetic linkage maps were constructed based on the smaller and bigger F2 populations, respectively. The candidate gene was mapped in an interval of 50.97 kb between ND78-3 and ND78-4. CG, candidate gene.

**Figure 4 plants-12-03340-f004:**
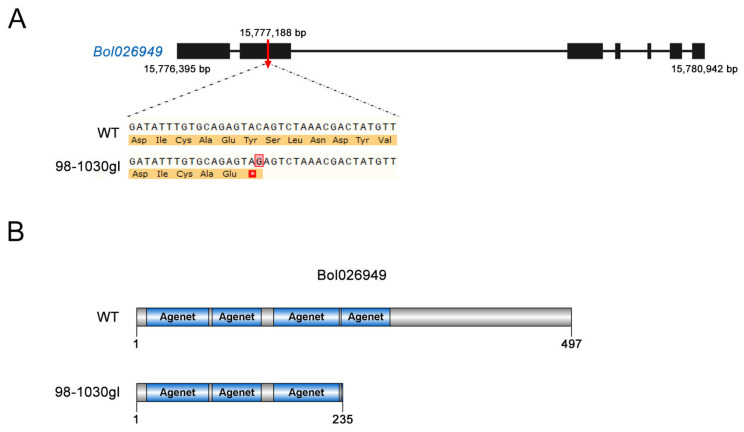
Mutation sites and amino acid changes in Bol026949. (**A**) Gene structure of Bol026949 and its mutation in 98-1030gl. The black boxes indicate exons, and the black lines represent introns. The red vertical arrow indicates the mutation position in 98-1030gl. The SNP mutation (C → G) resulted in the formation of a stop codon (TAG). (**B**) Schematic diagram showing amino acid differences between Bol026949 and its mutant.

**Figure 5 plants-12-03340-f005:**
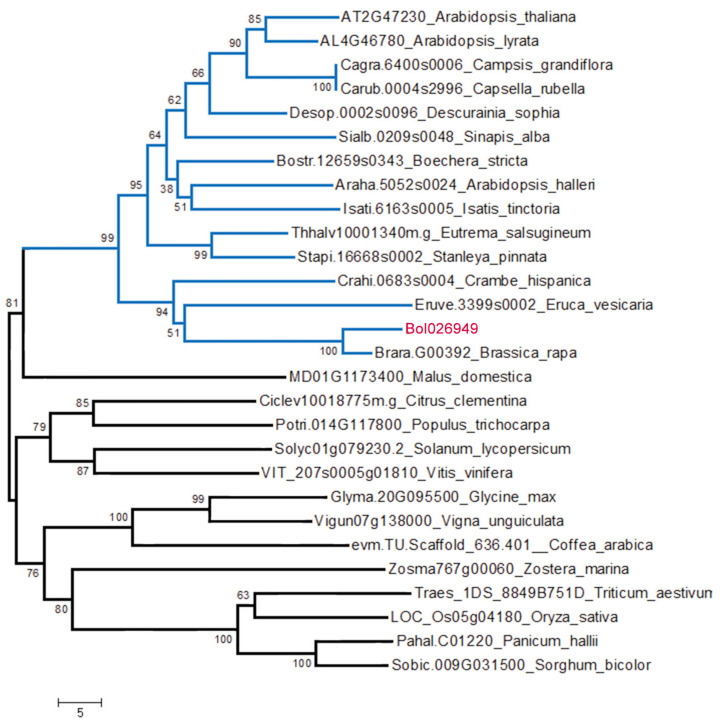
Phylogenetic analysis of Bol026949 and its homologous proteins.

**Figure 6 plants-12-03340-f006:**
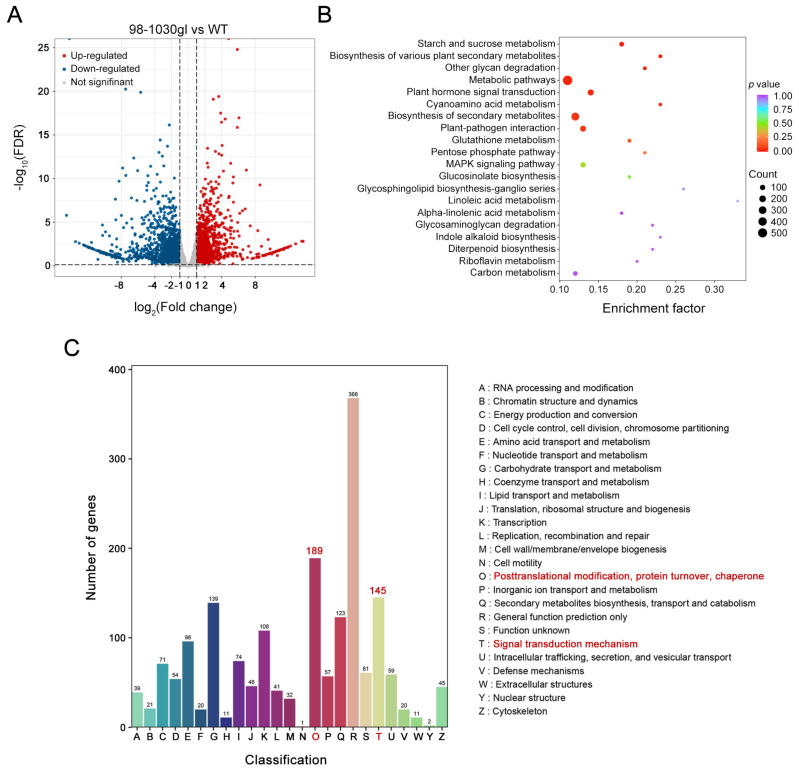
Differentially expressed genes between 98-1030gl and WT leaves. (**A**) Volcano plot showing differentially expressed genes in 98-1030gl relative to WT. (**B**) Kyoto Encyclopedia of Genes and Genomes pathway enrichment analysis of differentially expressed genes between 98-1030gl and WT. (**C**) Clusters of orthologous groups function enrichment analysis of differentially expressed genes between 98-1030gl and WT.

**Table 1 plants-12-03340-t001:** Inheritance analysis of the glossy green trait in a six-generation genetic population.

Population	Total	Glaucous	Glossy	Segregation Ratio	χ^2^	*p* Value
P1 (98-1030)	30	30	0	—	—	
P2 (98-1030gl)	30	0	30	—	—	
F1	132	132	0	—	—	
F2	875	650	225	2.89:1	0.22	0.64
BC1 (F1 × 98-1030)	204	204	0	—	—	
BC2 (F1 × 98-1030gl)	168	92	76	1.21:1	1.52	0.22

**Table 2 plants-12-03340-t002:** Potential candidate genes in the target mapping region.

Gene ID	Start Position	Stop Position	DNA Strand	Homologous Gene in Arabidopsis	Function Annotation
Bol026947	15767582	15769050	reverse	AT1G54680.4	unknown protein
Bol026948	15773097	15774439	reverse	—	unknown protein
Bol026949	15776395	15780942	forward	AT1G26540	Agenet domain-containing protein-related
Bol026950	15803605	15804112	reverse	AT1G20340	plastocyanin (petE)
Bol026952	15815490	15817138	reverse	AT1G20370	tRNA pseudouridylate synthase

## Data Availability

The data supporting the findings of this study are included in this article and its Appendix A. All data that support the findings of this study are available from the corresponding author upon request.

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
