# Peer review of "Fine Mapping and Identification of a Candidate Gene for the Glossy Green Trait in Cabbage (Brassica oleracea var. capitata)"

_plants, 2023, doi:10.3390/plants12183340_

Round 1

Reviewer 1 Report

Dear Authors,

I revised the manuscript number ID Plants 2588440 titled: “Fine mapping and identification of a candidate gene for the glossy green trait in cabbage (Brassica oleracea var. capitata)”.

The Authors reported the identification of a putative single recessive gene in the cabbage glossy green mutant 98-1030gl.

To my opinion, the introduction, the discussion and the results are consistent and useful for publication.

I suggest a few revision:

- page 3, line 98: specify how many times the Authors selfed the glossy green plants to reach homozygous glossy green mutant population (homozygous progeny),

-  page 6, line 179: explain more in detail about the annotation and sequence analysis results to declare that Bol026949 is the candidate gene for glossy green trait

Best regards

Reviewer 2 Report

the manuscript presents the results of research on identifying candidate genes for the glossy green trait in cabbage. the experience is well planned, the research methods are well selected and the results obtained are of great cognitive and practical importance. In my opinion, the work needs a few corrections before publication. in the description of the research material, the authors write about 6 groups/populations analyzed in genetic studies and mapping, but in fact they only use the parent plants and the F2 population. the remaining plants were used only to assess the inheritance of the analyzed feature - this should be clarified. What was the probability value for the ch2 tests? it should be given in the text or in the table. In conclusion: the authors write that they have identified the mutants - but the paper clearly states that these mutants have been identified earlier - so the conclusions should be changed or information should be added so that they have been identified in earlier studies. the present statement is misleading.
